# Intrathecal Actions of the Cannabis Constituents Δ(9)-Tetrahydrocannabinol and Cannabidiol in a Mouse Neuropathic Pain Model

**DOI:** 10.3390/ijms23158649

**Published:** 2022-08-03

**Authors:** Sherelle L. Casey, Vanessa A. Mitchell, Eddy E. Sokolaj, Bryony L. Winters, Christopher W. Vaughan

**Affiliations:** Pain Management Research Institute, Kolling Institute, University of Sydney at Royal North Shore Hospital, Sydney, NSW 2065, Australia; sherelle.casey@gmail.com (S.L.C.); vanessa.mitchell@sydney.edu.au (V.A.M.); esok5506@uni.sydney.edu.au (E.E.S.); bryony.winters@sydney.edu.au (B.L.W.)

**Keywords:** cannabinoid, neuropathic pain, THC, cannabidiol, synergy, intrathecal, mice

## Abstract

(1) Background: The psychoactive and non-psychoactive constituents of cannabis, Δ9-tetrahydrocannabinol (THC) and cannabidiol (CBD), synergistically reduce allodynia in various animal models of neuropathic pain. Unfortunately, THC-containing drugs also produce substantial side-effects when administered systemically. We examined the effectiveness of targeted spinal delivery of these cannabis constituents, alone and in combination. (2) Methods: The effect of acute intrathecal drug delivery on allodynia and common cannabinoid-like side-effects was examined in a mouse chronic constriction injury (CCI) model of neuropathic pain. (3) Results: intrathecal THC and CBD produced dose-dependent reductions in mechanical and cold allodynia. In a 1:1 combination, they synergistically reduced mechanical and cold allodynia, with a two-fold increase in potency compared to their predicted additive effect. Neither THC, CBD nor combination THC:CBD produced any cannabis-like side-effects at equivalent doses. The anti-allodynic effects of THC were abolished and partly reduced by cannabinoid CB1 and CB2 receptor antagonists AM281 and AM630, respectively. The anti-allodynic effects of CBD were partly reduced by AM630. (4) Conclusions: these findings indicate that intrathecal THC and CBD, individually and in combination, could provide a safe and effective treatment for nerve injury induced neuropathic pain.

## 1. Introduction

Chronic neuropathic pain is a debilitating pain syndrome caused by central or peripheral nervous system lesions and disease [1]. It is a difficult pain condition to manage, with many patients experiencing ongoing pain which is refractory to currently available pharmacotherapies [2]. These therapies are further limited by side effects, which often render them intolerable. Consequently, there is a need for alternative front-line and adjuvant therapeutics. Extracts from the plant Cannabis sativa are thought to have potential in treating several conditions such as pain [3]. Cannabis contains hundreds of phytocannabinoids, including the major psychoactive constituent Δ9-tetrahydrocannabinol (THC), and other non-psychoactive constituents such as cannabidiol (CBD). While several clinical trials have shown that whole raw cannabis, THC or combinations of THC and CBD (nabiximols) have potential in the treatment of neuropathic pain, there are questions over their clinical efficacy and safety [4,5].

There is growing animal evidence that systemic delivery of THC and CBD reduces the allodynia associated with a range of neuropathic pain models induced by nerve injury, chemotherapeutic drugs and streptozotocin [6,7,8,9,10,11,12,13,14,15,16,17,18]. Interestingly, systemically injected THC and CBD synergistically reduce allodynia in neuropathic pain models [6,7]. However, the systemic actions of THC-containing phytocannabinoid preparations in neuropathic pain models are associated with substantial cannabinoid-like side-effects, when injected or administered orally [6,10,19,20,21]. These side-effect issues are reflected by the problems with adverse reactions reported in clinical studies which are largely focused on systemic administration [4,5]. One approach to avoiding side-effects associated with cannabis has been the use of non-psychoactive phytocannabinoid constituents such as CBD [6,13,21].

Another approach to lessening the side-effects associated with cannabinoids is to use site-directed drug delivery. Several studies have demonstrated that intrathecal delivery of synthetic cannabinoid CB1 and CB2 receptor agonists reduces allodynia in a range of neuropathic pain models of nerve injury, chemotherapeutic drugs, diabetes and cancer [22,23,24,25,26,27,28,29,30,31,32,33,34]. Surprisingly, relatively little is known about the intrathecal effects of the cannabis constituents, THC and CBD, in neuropathic pain models [35,36]. We therefore explored the individual and combined intrathecal actions of THC and CBD in a mouse model of neuropathic pain. This included an assessment of their anti-allodynic actions and side-effects to determine their pain-relieving efficacy and safety.

## 2. Results

### 2.1. Time Course of Action of Intrathecally Administered THC and CBD

We first examined the time course of effect of acute intrathecal delivery of THC and CBD at a near maximal dose (100 nmol). There was a significant effect of both drug treatment and time for mechanical PWT (two-way ANOVA main effects: F(2, 15) = 4.5, *p* = 0.03; F(5, 75) = 58.3, *p* < 0.0001; interaction: F(10, 75) = 3.3, *p* = 0.001). Mechanical PWT was significantly less at 10–12 days following CCI surgery compared to pre-surgery levels, for all three treatment groups (Figure 1A, *p* < 0.0001, Sidak post hoc comparisons). At 10–12 days post-CCI surgery, intrathecal THC produced a significant increase in mechanical PWT at 1–4 h post-injection compared to the pre-injection baseline level (Figure 1A, *p* < 0.0001, *p* < 0.0001 and *p* = 0.007 for pre- versus 1, 2 and 4 h post-injection, Sidak post hoc comparisons). Similarly, intrathecal CBD produced a significant increase in mechanical PWT at 1–2 h post-injection compared to the pre-injection baseline level (Figure 1A, *p* = 0.0004, *p* < 0.0001 for pre- versus 1 and 2 h post-injection, Sidak post hoc comparisons). By contrast, mechanical PWT at all time points following injection of vehicle did not differ to the pre-injection baseline level (Figure 1A,B, *p* > 0.05 for pre- versus 1–6 h post-injection, Sidak post hoc comparisons).

There was a significant effect of both drug treatment and time for acetone responses (two-way ANOVA main effects: F(2, 15) = 8.8, *p* = 0.003, F(5, 75) = 45.3, *p* < 0.0001; interaction: F(10, 75) = 4.4, *p* < 0.0001). The number of acetone responses was significantly greater at 10–12 days following CCI surgery compared to pre-surgery levels, for all three treatment groups (Figure 1B, *p* < 0.0001, Sidak post hoc comparisons). At 10–12 days post-CCI surgery, intrathecal THC produced a significant decrease in acetone responses at 1–4 h post-injection compared to the pre-injection baseline level (Figure 1B, *p* < 0.0001, *p* < 0.0001 and *p* = 0.0001 for pre- versus 1, 2 and 4 h post-injection, Sidak post hoc comparisons). Intrathecal CBD produced a significant decrease in acetone responses at 1–2 h post-injection compared to the pre-injection baseline level (Figure 1B, *p* = 0.003, *p* = 0.003 for pre- versus 1 and 2 h post-injection, Sidak post hoc comparisons). By contrast, the number of acetone responses at any time point following injection of vehicle did not differ to the pre-injection baseline level (Figure 1B, *p* > 0.05 for pre- versus 1–6 h post-injection, Sidak post hoc comparisons).

In these animals there was a significant effect of time, but not drug treatment for rotarod latency (two-way ANOVA main effects: F(5, 75) = 26.4, *p* < 0.0001; F(2,15) = 1.0, *p* > 0.05; interaction: F(10,75) = 0.8, *p* > 0.05), but there was no significant effect of time, or drug treatment for bar latency (two-way ANOVA main effects: F(5, 75) = 1.4, *p* > 0.05; F(2,15) = 2.5, *p* > 0.05; interaction: F(10,75) = 0.8, *p* > 0.05). Rotarod latency, but not bar latency was significantly less at 10–12 days following CCI surgery compared to pre-surgery levels, for all three treatment groups (Figure 1C,D, rotarod: *p* < 0.0001; bar test: *p* > 0.05, Sidak post hoc comparisons). At 10–12 days post-CCI surgery, neither rotarod nor bar latency differed at any time point following intrathecal injection of THC, CBD, or vehicle compared to the pre-injection baseline levels (Figure 1C,D, *p* > 0.05; for pre- versus 1–6 h post-injection, Sidak post hoc comparisons). Subsequent dose–response analysis was therefore at 1–2 h after drug administration which covered the time of peak anti-allodynic effect.

### 2.2. Dose–Response Profiles of Intrathecal THC and CBD

We next examined the effect of a range of doses of intrathecal THC and CBD (1–178 nmol). Both THC and CBD produced a highly efficacious dose-dependent reversal of the CCI induced reduction in mechanical PWT, with ED_50_s of 14 and 21 nmol, respectively (Figure 2A, Table 1). In addition, both THC and CBD produced a moderately efficacious dose-dependent reduction in the CCI induced increase in acetone responses, with ED_50_s of 20 an 11 nmol, respectively (Figure 2B, Table 1).

Dose–response analysis was not performed for the side-effect measure because there was no significant effect of drug treatment or dose for rotarod latency (two-way ANOVA main effects: F(1, 60) = 0.2, *p* > 0.05, F(5, 60) = 0.1, *p* > 0.05), bar latency (two-way ANOVA main effects: F(1, 60) = 3.5, *p* > 0.05, F(5, 60) = 2.4, *p* > 0.05), or open field crossings (two-way ANOVA main effects: F(1, 60) = 1.0, *p* > 0.05, F(5, 60) = 0.8, *p* > 0.05). Thus, neither THC nor CBD had dose dependent effects on rotarod latency, bar latency, or the number of open field crossings at any dose tested (Figure 2C–E).

### 2.3. Effect of a Fixed-Ratio Combination of THC and CBD Intrathecal

We next examined the effects of intrathecal delivery of combination THC:CBD. When averaged across both mechanical and cold allodynia assays, the ratio of the individual ED_50_s of THC and CBD was approximately 1:1 by weight. Combination THC:CBD was therefore administered at this fixed ratio. Combination THC:CBD produced a highly efficacious dose-dependent reduction in mechanical PWT, with an ED_50_ of 9 nmol (Figure 3A, Table 1). Combination THC:CBD also produced a moderately efficacious dose-dependent reduction in acetone responses, with an ED_50_ of 9 nmol (Figure 3B, Table 1). By contrast, combination THC:CBD did not have a dose dependent effect on rotarod latency, bar latency, or the number of open field crossings over all doses tested (Figure 3C–E, range = 1.5–153 nmol total dose for THC + CBD).

The effect of combination THC:CBD was analyzed using a non-linear isobolographic approach for its anti-allodynic effects (but not side-effects as none were observed). At the 1:1 fixed ratio, the experimentally obtained dose–response curves for the anti-allodynic effects of combination THC:CBD were leftward shifted compared to their predicted additive dose–response curves, (Figure 3A,B, *p* < 0.01 for mechanical PWT and *p* < 0.05 for acetone responses at 8.4 and 15.3 mg.kg^−1^ doses, respectively).

The ED_20_–ED_50_ isoboles obtained using Equation (3) were non-linear and, unlike the ED_50_ isobole obtained using standard linear analysis with Equation (4), did not reach the x-axis at higher effect levels (Figure 4A,B). The experimentally obtained ED_50_s of combination THC:CBD were 8.6 and 9.4 nmol for mechanical PWT and acetone responses (respectively) and these were significantly less than their non-linear predicted additive ED_50_s of 14 and 20 nmol, respectively (Figure 4A,B, Table 1, *p* < 0.05, 0.001, for mechanical PWT and acetone responses). Indeed, the experimentally obtained ED_50_s of combination THC:CBD for mechanical PWT and acetone responses were near equivalent to their predicted ED_20_ and ED_30_ isoboles (Figure 4A,B).

### 2.4. Role of Cannabinoid CB1 and CB2 Receptors

We finally examined the role of cannabinoid receptors in the actions of the phytocannabinoids, THC and CBD (100 nmol, maximal doses), by co-administering them with maximal doses of the CB1 and CB2 receptor antagonists AM281 and AM630 (30 nmol each) [10,26,37]. There was a significant effect of phytocannabinoid and antagonist treatment on mechanical PWT (two-way ANOVA main effects: F(2, 45) = 42.9, *p* < 0.0001, F(2, 45) = 11.8, *p* < 0.0001; interaction: F(4, 45) = 9.7 *p* < 0.0001). Both THC and CBD reduced the CCI-induced decrease in mechanical PWT (Figure 5A, *p* < 0.0001 THC + vehicle v vehicle + vehicle; *p* = 0.0001 CBD + vehicle v vehicle + vehicle, Sidak post hoc comparisons). The effect of THC on mechanical PWT was abolished by AM281, but was unaffected by AM630 (Figure 5A, *p* < 0.0001 THC + vehicle v THC + AM281 and *p* > 0.05 THC + AM281 v vehicle + AM281; *p* > 0.05 THC + vehicle v THC + AM630, Sidak post hoc comparisons). The effect of CBD on mechanical PWT was unaffected by AM281, but partly reduced by AM630 (Figure 5A, *p* > 0.05 CBD + vehicle v CBD + AM281; *p* < 0.05 CBD + vehicle v CBD + AM630 and *p* < 0.01 CDB + AM630 v vehicle + AM630, Sidak post hoc comparisons). AM281 and AM630 alone did not have a significant effect on mechanical PWT (Figure 5A, *p* > 0.05 for vehicle + vehicle v vehicle + AM281 and vehicle + AM630, Sidak post hoc comparisons, Sidak post hoc comparisons).

There was a significant effect of phytocannabinoid and antagonist treatment on acetone responses (two-way ANOVA main effects: F(2, 45) = 18.7, *p* < 0.0001, F(2, 45) = 8.4, *p* < 0.001; interaction: F(4, 45) = 15.1, *p* < 0.0001). Both THC and CBD reduced the CCI-induced increase in acetone responses (Figure 5B, *p* < 0.0001 THC + vehicle v vehicle + vehicle; *p* < 0.001 CBD + vehicle v vehicle + vehicle, Sidak post hoc comparisons). The effect of THC on acetone responses was abolished by AM281 and reduced by AM630 (Figure 5B, *p* < 0.0001 THC + vehicle v THC + AM281 and *p* > 0.05 THC + AM281 v vehicle + AM281; *p* < 0.01 THC + vehicle v THC + AM630 and *p* < 0.001 THC + AM630 v vehicle + AM630, Sidak post hoc comparisons). The effect of CBD on acetone responses was unaffected by AM281, but abolished by AM630 (Figure 5B, *p* > 0.05 CBD + vehicle v CBD + AM281; *p* < 0.05 CBD + vehicle v CBD + AM630, *p* > 0.05 CDB + AM630 v vehicle + AM630, Sidak post hoc comparisons). AM281 and AM630 alone did not have a significant effect on acetone responses (Figure 5B, *p* > 0.05 for vehicle + vehicle v vehicle + AM281 and vehicle + AM630, Sidak post hoc comparisons, Sidak post hoc comparisons).

## 3. Discussion

In the present study it has been demonstrated that intrathecal delivery of the phytocannabinoids THC and CBD reduces allodynia in a nerve injury induced model of neuropathic pain. In combination, these phytocannabinoids acted synergistically to reduce allodynia. Furthermore, THC and CBD, alone and in combination was not associated with the adverse side-effects commonly associated with systemic cannabis administration. These findings indicate that the phytocannabinoids THC and CBD act synergistically within the spinal cord to reduce neuropathic pain and may therefore have benefit in the treatment of this intractable condition.

### 3.1. Spinally Delivered Phytocannabinoids

In the present study it has been shown that intrathecal delivery of both THC and CBD reduce mechanical and cold allodynia in a nerve injury induced neuropathic pain model. While this is the first study to examine the spinal anti-allodynic actions of the phytocannabinoids THC and CBD, these anti-allodynic actions are consistent with those previously reported for synthetic cannabinoid receptors agonists and dihydroxyl phytocannabinoid analogues in a range of neuropathic pain models [22,23,24,25,26,27,28,29,30,31,32,33,34,35].

While both THC and CBD were highly efficacious against mechanical allodynia, they had only partial effectiveness against cold allodynia. This differs to systemic administration where THC has high efficacy, compared to CBD which has only partial efficacy against both mechanical and cold allodynia [6,21]. This difference may be due a relatively greater role of spinal pain pathways in the suppressive actions of THC and CBD on mechanical allodynia, compared to cold allodynia. It also indicates that a single measure of neuropathic pain might not be a reliable indicator of drug efficacy against the range of abnormal signs associated with neuropathic pain models [8,10,38].

Interestingly, both THC and CBD produced no cannabinoid-like side-effects when delivered spinally, even at doses which were 8–15 times greater than their anti-allodynia ED_50_s. This differs to prior studies in which intrathecal delivery of pan-cannabinoid receptor agonists, such as HU210 and CP55940, reduce allodynia and produce cannabinoid side-effects at similar doses [23,29]. The difference between THC and selective agonists may be due to the lower affinity and efficacy of THC for cannabinoid receptors, or other experimental factors (see Section 3.3). The lack of side-effects of intrathecal THC and CBD might also be contrasted to systemic THC which has a relatively poor therapeutic window [6,21]. This is consistent with prior studies that have demonstrated a major role of the brain in the side-effects of THC.

### 3.2. Spinally Delivered Phytocannabinoids in Combination

Intrathecal administration of combination THC:CBD also reduced mechanical and cold allodynia. Non-linear isobolographic analysis indicated that this anti-allodynia was synergistic, with combination THC:CBD having a two-fold greater potency compared to its predicted additive effect. This synergistic interaction following spinal delivery is consistent with recent isobolographic studies on systemically administered phytocannabinoids [6,7]. Furthermore, combination THC:CBD did not produce any cannabinoid-like side-effects, even at doses that were 20 times greater than their anti-allodynia ED_50_s. These findings indicate that intrathecally delivered THC and CBD, alone and in combination, are relatively safe, at least in terms of the acute side-effects commonly observed with cannabis.

### 3.3. Role of Cannabinoid Receptors

The role of cannabinoid CB1 and CB2 receptors differed between the two phytocannabinoids. The THC induced reduction in mechanical allodynia was abolished by the cannabinoid CB1 receptor antagonist AM281, but unaffected by the cannabinoid CB2 receptor antagonist AM630. This was consistent with prior intrathecal neuropathic studies using cannabinoid CB1 selective receptor agonists, antagonists and knockdown of CB1 receptors [23,28,31,32]. By contrast, the THC induced reduction in cold allodynia was abolished by AM281 and partly reduced by AM630. This was consistent with several studies using cannabinoid CB2 selective agonists, antagonists and/or knockout [24,25,26,27,29,33]. These observations suggest that CB2 receptors have a greater role in the spinal processing of cold allodynia compared to mechanical allodynia. This difference between the role of CB1 and CB2 receptors may have been due to a number of factors including the relatively low affinity and efficacy of THC for cannabinoid receptors compared to synthetic agonists [39]. It is also possible that the varying roles of CB1 and CB2 receptors may be related to differing experimental factors in the present versus prior studies, including species (rats versus mice), the type of chronic pain model (CCI versus other forms of nerve injury, streptozotocin-induced diabetes, chemotherapy drugs and bone cancer) and even the pain assays used (mechanical/cold allodynia versus spontaneous pain and thermal hyperalgesia).

The CBD induced reduction in mechanical and cold allodynia was unaffected by AM281 which is consistent with its low affinity for cannabinoid CB1 receptors [39]. By contrast, the CBD induced decrease in mechanical and cold allodynia was partly reduced by AM630. This is consistent with the known actions CB2 selective agonists (see above) and the moderate affinity and efficacy of CBD for cannabinoid CB2 receptors [39]. The partial involvement of cannabinoid CB2 receptors indicates that other systems within the spinal cord are important neuropathic pain targets for CBD, such as serotonergic 5HT1A receptors [13]. It should also be noted that intrathecal administration of the CBD analogue dihydroxyl-cannabidiol, which has low affinity for cannabinoid receptors compared to CBD, reduces allodynia by allosteric modulation of glycine receptors [35,40]. Thus, CBD is likely to reduce allodynia via multiple targets.

### 3.4. Conclusions

The present findings indicate that intrathecal delivery of the phytocannabinoids THC and CBD reduces the mechanical and cold allodynia associated with a nerve injury induced model of neuropathic pain. Interestingly, THC and CBD acted synergistically to reduce allodynia, leading to a substantial increase in their anti-allodynic potency. In addition, both THC and CBD were devoid of the cannabis-like side-effects associated with the systemic delivery of THC-containing cannabinoids. These findings indicate that spinal delivery of the primary phytocannabinoids of the plant Cannabis sativa has potential in the treatment of chronic neuropathic pain.

## 4. Materials and Methods

All experiments in this study were carried out on 8–12-week-old male C57BL/6 mice obtained from the Kolling Institute Animal Facility. All data are reported in compliance with the ARRIVE guidelines and those of the ‘NH&MRC Code of Practice for the Care and Use of Animals in Research in Australia’. Mice were initially housed in groups of 4 littermates, and then individually in adjacent cages following surgery. Individually ventilated cages were maintained at 22–23 °C and humidity 65–75%, with a 12:12 h light: dark cycle. Animals had ad libitum access to food and water throughout all stages of the study. Cages were enriched with a mouse house igloo, tissues for nesting and either a straw or paddle pop stick.

### 4.1. Neuropathic Pain Model

The chronic constriction injury (CCI) model, a commonly used neuropathic pain model, was used in this study [38]. Mice were anesthetized (2% isoflurane in saturated oxygen) and positioned on a heat mat to avoid hypothermia. Using an aseptic approach, the left common sciatic nerve was exposed, and two 6–0 chromic gut loose ligatures were placed 2 mm apart around the common sciatic nerve proximal to its trifurcation. The ligatures were lightly tightened until a twitch of the foot was observed, taking care not to compromise the blood flow to the nerve. The muscle over the nerve was then closed with 6–0 silk and the skin incision closed using tissue glue. The mice were monitored for recovery from anesthesia before being returned to their home cages and were then monitored daily until the day of the experiment.

### 4.2. Behavioural Testing

Nerve injury induced mechanical and cold allodynia were tested using plantar application of von Frey hairs and acetone, respectively. Animals were placed in an elevated Perspex chamber with a mesh wire floor and left to acclimatize for 30–60 min before any testing was conducted. Mechanical allodynia was tested by applying a series of von Frey filaments (0.2–6.84 g; North Coast Medical, San Jose, CA, USA) to the plantar surface of the operated hind paw. The mechanical paw withdrawal threshold (PWT) was calculated using the simplified up-down protocol [41]. A positive pain-like response was recorded as a rapid withdrawal, flinching, shaking, or licking of the paw. Cold allodynia was assessed by applying 20 µL of acetone to the operated hind paw to induce evaporative cooling. The number of pain-like responses was counted over a 2 min period.

Common cannabinoid side effects including motor impairment, catalepsy and sedation were also assessed. Motor impairment was tested using the rotarod, an assay in which mice are placed on a bar that slowly increases in speed from 4 to 30 rpm over a 300 s period. The time at which each mouse either fell off the bar or held on for 2 or more consecutive rotations was recorded (cut-off = 300 s). Catalepsy was assessed using the bar test (enclosure size 25 × 15 × 30 cm). Mice were placed with their forepaws on a raised (2 cm high) bar and hindpaws on the enclosure floor. The time taken to move from that position was recorded (cut-off = 120 s). Sedation was measured using the dark open field test. Mice were placed in a novel environment (25 cm × 25 cm open-topped Perspex box) and a 4 × 4 square grid super-imposed over the recording to count the number of times the hindquarters of the animal crossed a line on the grid. All behavioral tests were performed under low-level red light (3–4 lx).

### 4.3. Experimental Protocol

Animals were initially habituated to experimenter handling, the allodynia testing chambers and trained on the rotarod device before obtaining baseline allodynia/side-effect measurements. Animals then underwent CCI surgery, and drug testing was conducted at 10–12 days post-surgery. Each animal underwent only one drug testing experiment, and all animals were euthanized by carbon dioxide asphyxiation at the conclusion of the experiment. The experimenter was blinded to the drug being tested until data were collated.

For the time course experiments, allodynia and side-effect (except open field) measurements were taken immediately prior to drug administration, and then at 1, 2, 4 and 6 h post- drug administration. For the dose–response and antagonist experiments, allodynia and side-effect (except open field) measurements were taken immediately prior to drug administration and then at 1 and 2 h post- drug administration; time points which coincided with the time of peak drug effect determined in the time course experiments (see Figure 1). To maintain novelty, the open field test was performed only once at 1.5 h post-drug administration. The acetone, rotarod and bar tests were performed twice at each time point, and the von Frey and open field tests were performed only once at each time point.

### 4.4. Drugs and Administration

The phytocannabinoids THC and CBD were obtained from THCPharm (Frankfurt, Germany), AM281 and AM630 were from Cayman Chemicals (Ann Arbor, MI, USA). Stock solutions of all drugs were prepared in dimethyl sulfoxide. Drug injection solutions were made up immediately prior to administration. Drugs were administered as an intrathecal injection using a Hamilton syringe and 25 G needle (volume = 10 µL in a vehicle consisting of 25% DMSO, 15% ethanol and 2% randomly methylated beta-cyclodextrin (RAMEB) in saline). Intrathecal injections were made under brief anesthesia (2% Isoflurane in saturated oxygen) and recovered immediately following drug administration. For intrathecal injection, mice were placed in dorsal recumbency and the fur over the lumbar-sacral spine was clipped, and the skin wiped with ethanol. The needle was inserted in the intervertebral gap between L4 and L5 until a ‘popping’ sensation and/or twitch of the tail was observed, and the drug solution slowly injected. The needle was then slowly removed, and the animal placed in its home cage for recovery.

### 4.5. Analysis and Statistics

Data were analyzed using SPSS (ver. 26, IBM Corp), Excel and Prism (ver. 8, GraphPad Software). There were six animals in each treatment group and no animals were excluded from analysis. For the time course experiments, raw data were analyzed. For the dose–response and antagonist experiments, all data (except for open field) were normalized as a percentage of the maximum possible effect (MPE). For mechanical PWT, this was calculated as (post-drug − pre-drug)/(cut-off − pre-drug), with a cut-off of 6.84 g. For acetone and rotarod this was calculated as (pre-drug − post-drug)/(pre-drug). For the bar test this was calculated as (post-drug − pre-drug)/(cut-off), with a cut-off of 120 s. Where appropriate, data satisfied the Shapiro–Wilk test for normality, Mauchly’s test of sphericity (Greenhouse–Geisser Correction was applied if appropriate) and Levene’s test of equality of variance. Data are presented as mean ± SEM and considered significantly different when *p* < 0.05.

For the time course experiments, raw data were analyzed using two-way repeated ANOVA, with time and drug treatment as within- and between-subjects factors. Post hoc comparisons to the pre-injection time point were made using the Sidak correction. For the antagonist experiments, normalized data were analyzed using two-way ANOVA, with phytocannabinoid and antagonist treatment as between-subjects factors. Post hoc comparisons to the vehicle/vehicle and vehicle/antagonist treatment groups were made using the Sidak correction.

For dose–response experiments, normalized data were analyzed using two-way ANOVA, with phytocannabinoid and dose as between-subjects factors. The individual effects of THC and CBD, and combination THC:CBD were then fit with a sigmoidal function:Effect(Dose) = E_max_ × [1/(1 + 10^p(Log(ED50)-Log(Dose))^)],(1)
with a maximal effect of E_max_, a half-maximally effective dose of ED_50_ and a Hill slope of p. An isobolographic approach was used to determine whether there was an interaction between the effect of combination treatment with THC and CBD on allodynia, using a fixed ratio design [6,42]. To do this, the THC:CBD combination was examined at a 1:1 ratio by weight, which corresponded approximately to a 1:1 ratio by ED_50_. The dose–response curves for the predicted additive effect of combination THC and CBD were modelled using a non-linear isobolographic approach, which allows for differential maximal effects and Hill slopes, as follows:Effect(a,b) = [E_B_ × (b + {C_B_/k^1/p^})^p^]/[(b + (C_B_/k^1/p^)^p^ + C_B_^p^] where k = [(E_A_/E_B_) × (1 + C_A_^q^/a^q^)] − 1,(2)
at doses a and b for drugs A (CBD) and B (THC), with maximal effects of E_A_ and E_B_ (where E_B_ > E_A_), ED_50_s of C_A_ and C_B_ and Hill slopes of p and q, respectively. The standard errors of the ED_50_s for the predicted additive combination dose–responses curves were obtained using the delta method, with Taylor series approximation of equation (2). The predicted and experimental ED_50_s for the THC:CBD combinations were compared using an unpaired *t*-test [42]. The non-linear isoboles which describe the relationships between the drugs at doses a and b (for CBD and THC, respectively), at specified effect levels B_i_ (20–50% effect levels), were calculated using:b = B_i_ − [C_B_/({E_B_/E_A_} × {1 + (C_A_^q^/a^q^)} − 1)^1/p^)],(3)

This was also compared to the simple linear (50% effect level) isobole where drugs A and B have maximal effects of 100% and Hill slopes of unity,
b = E_B_ − [a × (E_B_/E_A_)],(4)

## Figures and Tables

**Figure 1 ijms-23-08649-f001:**
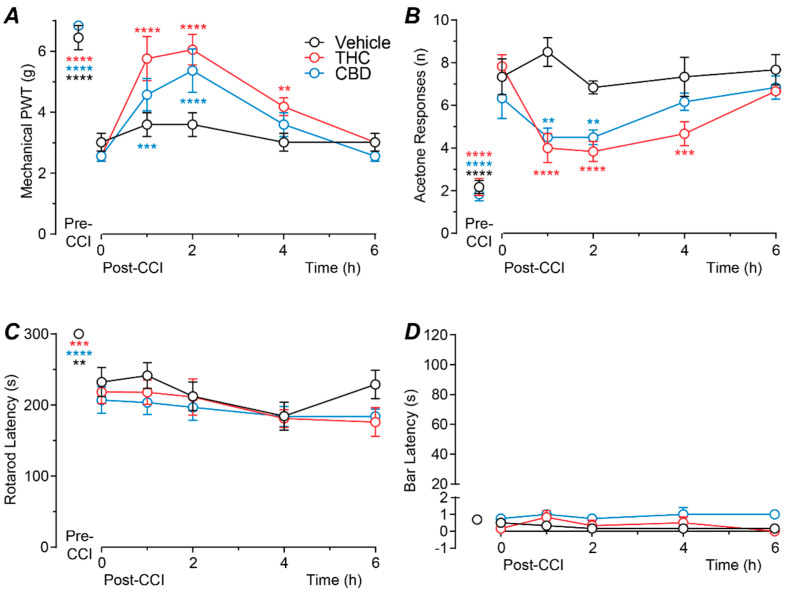
Time course of action of intrathecal THC and CBD. Time plots of the effects of intrathecally injected THC (100 nmol), CBD (100 nmol), or matched vehicle on (**A**) mechanical paw withdrawal threshold (PWT), (**B**) acetone responses, (**C**) rotarod latency and (**D**) bar latency (n = 6 per treatment group). Animals received a single intrathecal administration at time 0 h, 10-12 days after CCI surgery (post-CCI); pre-CCI data are also displayed. **, *** and **** denote *p* < 0.01, 0.001 and 0.0001 compared to time 0 h. within each treatment group. Data are shown as the mean ± s.e.mean of raw values.

**Figure 2 ijms-23-08649-f002:**
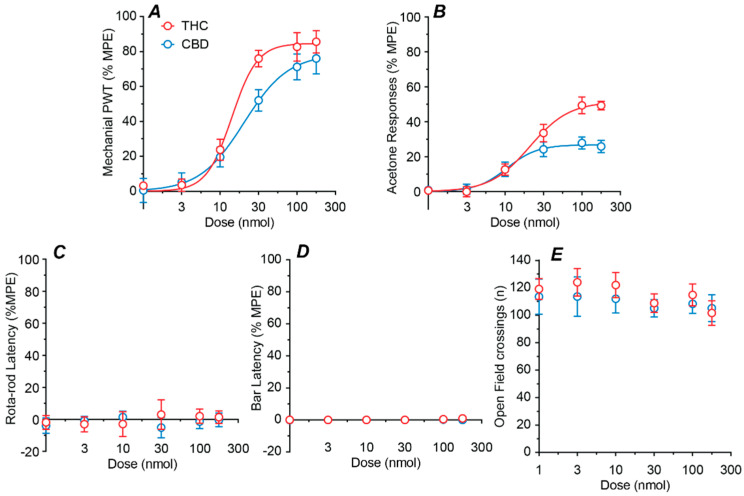
Dose–response curves for intrathecal THC and CBD. Dose–response curves for the effect of intrathecal THC and CBD on (**A**) mechanical paw withdrawal threshold (PWT), (**B**) acetone responses, (**C**) rotarod latency, (**D**) bar latency and (**E**) open field crossings. Where appropriate, the sigmoidal parametric fit is shown. All data are displayed as the mean ± s.e.mean percentage of the maximum possible effect (%MPE), except for open field (mean ± s.e.mean of raw data).

**Figure 3 ijms-23-08649-f003:**
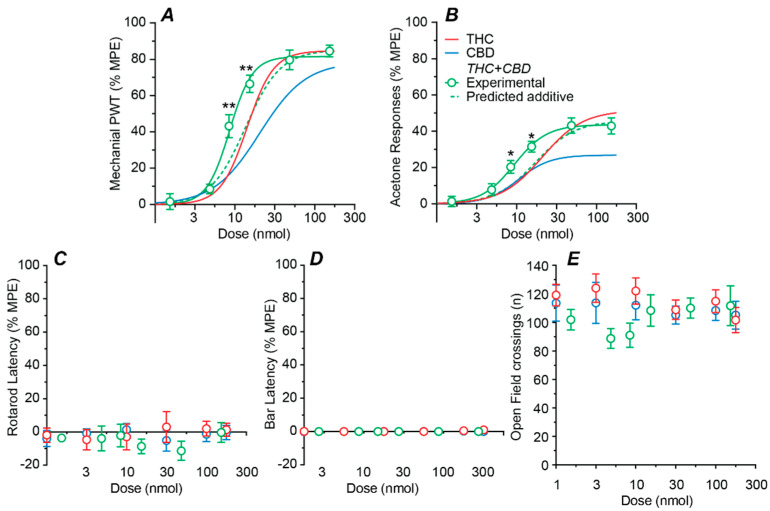
Dose–response curves for the effect of combined intrathecal THC and CBD. Dose–response curves showing the effect of administration of THC and CBD in a 1:1 fixed ratio on (**A**) mechanical paw withdrawal threshold (PWT), (**B**) acetone-induced responses, (**C**) rotarod latency, (**D**) bar latency and (**E**) open field crossings. The non-linear curve fits to the experimental combination data (solid lines) and predicted additive effect (dotted lines), where appropriate. Also shown are the sigmoidal fits for THC and CBD alone (solid lines). All data are displayed as the mean ± s.e.mean percentage of the maximum possible effect (%MPE), except for open field (mean ± s.e.mean of raw data are shown). * and ** denote *p* < 0.05, 0.01 for experimental THC:CBD data points v the predicted additive value at the corresponding dose.

**Figure 4 ijms-23-08649-f004:**
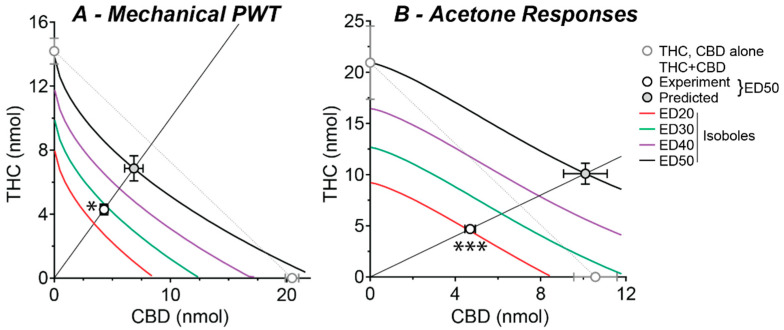
Isoboles for combined intrathecal THC and CBD treatment at a range of effect levels. Isoboles for the effect of intrathecal THC:CBD co-administration on (**A**) mechanical PWT and (**B**) acetone-induced responses, at a 1:1 fixed ratio combination. The experimental and predicted (Exp and Pred) ED_50_s are shown as part of the continuum of fixed-ratio effects. Theoretical isoboles of additivity for effect levels of 20, 30, 40 and 50 of maximum are shown (ED_20_–ED_50_, solid lines from Equation (3)); as a comparison the 50% effect level isobole is shown for the simple case where THC and CBD are assumed to have equal 100% maximal effects and Hill slopes of unity (dotted line). The individual ED_50_s for THC and CBD are shown on the x- and y-axes, respectively. * and *** denote *p* < 0.05 and 0.001 for the experimental versus predicted additive ED_50_.

**Figure 5 ijms-23-08649-f005:**
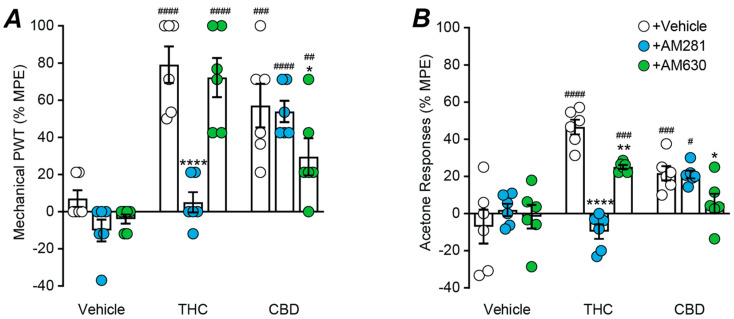
Effect of cannabinoid receptor antagonist on intrathecal THC and CBD induced anti-allodynia. Scatter plots of the effect of co-administration of the cannabinoid CB1 and CB2 receptor antagonists, AM281 and AM630 (30 nmol), on the effect of maximal intrathecal doses of THC (100 nmol), CBD (100 nmol) on (**A**) mechanical paw withdrawal threshold (PWT) and (**B**) acetone responses. The bars represent the mean ± s.e.mean of the percentage of the maximum possible effect (%MPE). *, **, **** denote *p* < 0.05, 0.01, 0.0001 for vehicle/THC/CBD + vehicle versus Vehicle/THC/CBD + AM281/AM630; #, ##, ###, #### denote *p* < 0.05, 0.01, 0.001, 0.0001 for vehicle + vehicle/AM281/AM630 versus THC/CBD + vehicle/AM281/AM630.

**Table 1 ijms-23-08649-t001:** Potency and efficacy of the intrathecally delivered phytocannabinoids THC and CBD, and the experimentally determined and predicted additive values for combination THC:CBD.

THC (Exp.)			CBD (Exp.)			THC + CBD	
						(Exp.)	(Pred.)
ED_50_	Hill Slope	E_max_	ED_50_	Hill Slope	E_max_	ED_50_	ED_50_
Mechanical PWT							
14 (0.8)	2.6 (0.3)	85 (2)	20 (0.6)	1.5 (0.1)	79 (2)	8.6 (0.7)	14 (1.6)
Acetone Responses							
21 (3.6)	1.7 (0.4)	52 (3)	11 (1.0)	2.2 (0.4)	27 (1)	9.4 (0.5)	20 (2.1)

Experimental (Exp.) and predicted (Pred.) data shown as mean (s.e.mean), with ED_50_ values in mg.kg^−1^, E_max_ values as % MPE.

## Data Availability

Not applicable.

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
