# Peer review of "Intrathecal Actions of the Cannabis Constituents Δ(9)-Tetrahydrocannabinol and Cannabidiol in a Mouse Neuropathic Pain Model"

_ijms, 2022, doi:10.3390/ijms23158649_

Round 1
Reviewer 1 Report
In this manuscript, the authors test the hypothesis that spinal cord restriction of cannabinoids (THC and CBD) may provide analgesic benefit without the adverse effects that THC has in the brain. They generate dose response curves for intrathecal injection of THC, CBD, and a 1:1 combination, and demonstrate that both can reverse CCI-induced pain responses to touch and cold, and that the mixture shows synergy via isobologram. Importantly, the authors report no effects on locomotor activity nor catalepsy. Finally, they show that both CB1 and 2 receptors may be contributing with THC acting more at CB1 and CBD at CB2. The authors conclude that spinally restricted administration of cannabinoids may be a viable treatment strategy for chronic pain.
This manuscript is well-written, and the conclusions drawn are justified by the results. Experiments are simply designed and well-analyzed. There are small issues noted but these should be relatively simple to respond to. Otherwise, this manuscript is suitable for publication.
Issues:
1) Body temperature is another part of the THC behavioral tetrad that could have been measured. Is there data on this?
2) It is surprising that the predicted additive line in Figure 3 closely overlaps the THC alone curve. I would think that if they were additive, there would be a small left-shift. In the isobologram for PWT in in Figure 4, it appears that the predicted ED50 (gray circle) sits below 8 (nmol, mg/kg? see below) but on the curve (green dashes) in Figure 3 the ED50 looks over 10, by eye. The authors should check the math on those and make sure they are congruent.
3) The axes in Figure 4 are labeled as mg/kg, but I’m pretty sure these are nmol injections. Please double check this and fix as appropriate.
4) Figure 3 legend: “solids” vs “solid” (line 155)
5) In figure 5, relatively low numbers of animals may be obscuring stronger effects. For example 1 animal seems to be driving up the CBD/AM630 score and that animal may be an outlier.
Author Response
Below are our responses to the comments of reviewer 1 on manuscript IJMS- 1816293. Changes to the manuscript have been highlighted in yellow (except minor typos as noted below).
1) Body temperature is another part of the THC behavioral tetrad that could have been measured. Is there data on this?
This was not assessed. We focused on two standard cannabinoid side-effects.
2) It is surprising that the predicted additive line in Figure 3 closely overlaps the THC alone curve. I would think that if they were additive, there would be a small left-shift. In the isobologram for PWT in in Figure 4, it appears that the predicted ED50 (gray circle) sits below 8 (nmol, mg/kg? see below) but on the curve (green dashes) in Figure 3 the ED50 looks over 10, by eye. The authors should check the math on those and make sure they are congruent.
In Fig 4A, the predicted additive ED50 is actually 14 nmol, as also indicated in table 1. This is the sum of the ED50 doses of THC and CBD = 6.9 + 6.9 nmol (from the x- and y-axes in Fig 4A).
3) The axes in Figure 4 are labeled as mg/kg, but I’m pretty sure these are nmol injections. Please double check this and fix as appropriate.
We thank the reviewer for pointing out this error. It has been corrected.
4) Figure 3 legend: “solids” vs “solid” (line 155)
Corrected
5) In figure 5, relatively low numbers of animals may be obscuring stronger effects. For example 1 animal seems to be driving up the CBD/AM630 score and that animal may be an outlier.
We have deliberately not excluded any animals as outliers. Our power analysis, based on similar prior studies we have performed, indicated that 6 animals per group was sufficient given the effects sizes we have encountered in these studies.
Reviewer 2 Report
The topic of this article is interesting, the authors presenting the effects of intrathecal administration of Δ(9)-tetrahydro- 2 cannabinol and cannabidiol in chronic sciatic nerve constriction model of neuropathic pain in mice.
After reading the manuscript, the following doubts and suggestions have arisen.
The introduction and the discussion sections should be more complete, providing supplementary background about the pharmacology of the cannabis constituents, and their effects in different type of neuropathic pain in laboratory animals, as well as their possible efficiency in patients (see:
· Hoot MR et al. Chronic constriction injury reduces cannabinoid receptor 1 activity in the rostral anterior cingulate cortex of mice. Brain Res 2010; 1339:18-25.
· Campos RMP et al. Cannabinoid Therapeutics in Chronic Neuropathic Pain: From Animal Research to Human Treatment. Front Physiol 2021;12:785176.
· Chang F-C et al. Optimizing the Synergistic Effects of Cannabidiol and Δ9-Tetrahydrocannabinol for the Treatment of Neuropathic Pain in Mouse Behavioural Models. URNCST Journal 2020; 4(6): 1-8.
· Wen J et al. WWL70 protects against chronic constriction injury-induced neuropathic pain in mice by cannabinoid receptor-independent mechanisms. J Neuroinflammation. 2018;15(1):9.
· Silva-Cardoso GK et al. Cannabidiol effectively reverses mechanical and thermal allodynia, hyperalgesia, and anxious behaviors in a neuropathic pain model: possible role of CB1 and TRPV1 receptors. Neuropharmacology 2021, 197:108712.
· Henderson-Redmond AN et al. Sex Differences in Tolerance to Delta-9-Tetrahydrocannabinol in Mice With Cisplatin-Evoked Chronic Neuropathic Pain. Front Mol Biosci. 2021 Jun 25;8:684115.
The results obtained should be compared with those achieved by other researchers and discussions should be significantly detailed. In discussion section, the authors need to develop argumentation in depth based on the current understanding and the findings of the results obtained, presenting the potential, the weakness and limitation, and future research direction, among others. Authors should try to explain the theoretical implication as well as the translational application of their research.
Some other aspects were found in this manuscript:
- different fonts were used in the text and in the figures;
- the authors should improve the quality of the figures
- the authors should upgrade the references;
- spelling check of the text is mandatory.
- English including grammar, style and syntax, should be improved through the professional help from English Editing Company for Scientific Writings.
- a schematic representation of the study would be appreciated.
Author Response
Below are our responses to the comments of reviewer 2 on manuscript IJMS- 1816293. Changes to the manuscript have been highlighted in yellow (except minor typos as noted below).
1) The introduction and the discussion sections should be more complete, providing supplementary background about the pharmacology of the cannabis constituents, and their effects in different type of neuropathic pain in laboratory animals, as well as their possible efficiency in patients (see:
- Hoot MR et al. Chronic constriction injury reduces cannabinoid receptor 1 activity in the rostral anterior cingulate cortex of mice. Brain Res 2010; 1339:18-25.
- Campos RMP et al. Cannabinoid Therapeutics in Chronic Neuropathic Pain: From Animal Research to Human Treatment. Front Physiol 2021;12:785176.
- Chang F-C et al. Optimizing the Synergistic Effects of Cannabidiol and Δ9-Tetrahydrocannabinol for the Treatment of Neuropathic Pain in Mouse Behavioural Models. URNCST Journal 2020; 4(6): 1-8.
- Wen J et al. WWL70 protects against chronic constriction injury-induced neuropathic pain in mice by cannabinoid receptor-independent mechanisms. J Neuroinflammation. 2018;15(1):9.
- Silva-Cardoso GK et al. Cannabidiol effectively reverses mechanical and thermal allodynia, hyperalgesia, and anxious behaviors in a neuropathic pain model: possible role of CB1 and TRPV1 receptors. Neuropharmacology 2021, 197:108712.
- Henderson-Redmond AN et al. Sex Differences in Tolerance to Delta-9-Tetrahydrocannabinol in Mice With Cisplatin-Evoked Chronic Neuropathic Pain. Front Mol Biosci. 2021 Jun 25;8:684115.
We have already included reference to 16 original research papers which have studied the systemic effects of two main cannabis constituents THC and CBD in a range of neuropathic pain models (refs 6 – 21), and 15 original research papers which have studied the spinal effects of cannabinoids. In doing so, we have focused on the studies which first reported their actions in the various pain models.
We believe that adding to this extensive list of reference studies is unnecessary. In addition, the crucial information is that there are relatively few studies on the spinal actions of the phytocannabinoids THC and CBD in a neuropathic pain model, and none about their effect in combination. We do not see the value in referencing additional papers, particularly those which have studied actions in specific brain regions (e.g. Hoot) as they are unrelated to their spinal actions.
2) The results obtained should be compared with those achieved by other researchers and discussions should be significantly detailed. In discussion section, the authors need to develop argumentation in depth based on the current understanding and the findings of the results obtained, presenting the potential, the weakness and limitation, and future research direction, among others. Authors should try to explain the theoretical implication as well as the translational application of their research.
This is already discussed in the manuscript
3) Some other aspects were found in this manuscript:
- different fonts were used in the text and in the figures;
- the authors should improve the quality of the figures
- the authors should upgrade the references;
- spelling check of the text is mandatory.
We believe (i) we have followed the formatting guidelines, (ii) that the figures are of high quality, (iii) the references are adequate (see Qu1), (iv) spelling is correct.
4) English including grammar, style and syntax, should be improved through the professional help from English Editing Company for Scientific Writings.
We have rechecked the grammar & style and cannot find any issues.
5) a schematic representation of the study would be appreciated.
We do not think this would add much to the manuscript, but can add something if the editors deem it helpful.